# From global to local MDI variable importances for random forests and when they are Shapley values

**Antonio Sutera,** [*] **Gilles Louppe, Van Anh Huynh-Thu, Louis Wehenkel, Pierre Geurts**
Dept. of EE & CS, University of Liège, Belgium
{a.sutera,g.louppe,vahuynh,l.wehenkel,p.geurts}@uliege.be

## Abstract

Random forests have been widely used for their ability to provide so-called *importance measures*, which give insight at a global (per dataset) level on the relevance of input variables to predict a certain output. On the other hand, methods based on Shapley values have been introduced to refine the analysis of feature relevance in tree-based models to a local (per instance) level. In this context, we first show that the global Mean Decrease of Impurity (MDI) variable importance scores correspond to Shapley values under some conditions. Then, we derive a *local* MDI importance measure of variable relevance, which has a very natural connection with the global MDI measure and can be related to a new notion of local feature relevance. We further link local MDI importances with Shapley values and discuss them in the light of related measures from the literature. The measures are illustrated through experiments on several classification problems.

## 1 Motivation

While research in machine learning (ML) often focuses on predictive accuracy, another important topic concerns the interpretation of ML models and their predictions. Interpreting a model helps to uncover the mechanisms it captures (*e.g.*, biomarkers useful to diagnose a disease), and to explain its predictions (*e.g.*, why a particular patient is diagnosed healthy or sick). The latter becomes essential when a ML prediction may impact one's life, and as a way of checking the trustworthiness of a ML model (*e.g.*, to identify unwanted biases). Common interpretation tools include variable importance measures that assess which, and to which extent, variables are important for a model. They help to understand how the model works, and to gain insight on the underlying modelled mechanism.

In tree-based methods, such as Random forests [Breiman, 2001], feature importance scores can be derived as a low-cost by-product of the learning step. Given their extensive use in applied research, tree-based importance measures have been studied both empirically (see, *e.g.*, Strobl et al. [2007], Archer and Kimes [2008], Genuer et al. [2010], Auret and Aldrich [2011]) and theoretically (see, *e.g.*, Ishwaran et al. [2007], Louppe et al. [2013], Louppe [2014], Sutera et al. [2018], Li et al. [2019], Sutera [2019], Scornet [2020]). Assuming a sufficiently large learning set and number of trees, these works showed that importance measures have desirable properties, such as consistency with respect to the notion of feature relevance. They also analyzed the impact of learning meta-parameters (*e.g.*, randomization level, tree depth, ensemble size) on such properties. While standard importance measures evaluate the *global* importance of a feature at the level of a dataset, several works proposed new approaches based on Shapley values to derive *local* scores reflecting the importance of a feature for a given prediction (*e.g.*, Neto and Paulovich [2020], Lundberg et al. [2020], Izza et al. [2020]).

The contribution of the present work in this context is two-fold. First, we show that the standard mean decrease of impurity (MDI) measure when derived from totally randomized trees and in asymptotic

---

[*] Corresponding author. Email address: sutera.antonio@gmail.com

conditions (similar to those used to show the consistency with respect to the relevance) are Shapley values, and therefore have the same properties as any other importance measures based on these values (Section 3). Secondly, we propose a new local MDI measure for tree-based ensemble models to measure feature relevance locally, which naturally derives from global MDI and corresponds to Shapley values in the same conditions (Section 4). Global and local MDI are compared against other Shapley-value based scores, both conceptually (Section 5) and empirically (Section 6).

## 2 Background

In what follows, we consider a standard supervised learning setting and denote by $V = \{X_1, \ldots, X_p\}$ the set of $p$ input variables, and by $Y$ the output.

**Game theory and Shapley value.** We only remind here concepts and results that are useful later in the paper. Notations below are mostly adapted from [Besner, 2019].

In game theory, a *TU-game* $(V, v)$ (*i.e.*, *a cooperative game* with transferable utilities) is defined by a finite set of players $V = \{X_1, \ldots, X_p\}^2$ and a *characteristic (or coalition) function* $v \in \mathbb{V} : 2^V \to \mathbb{R}$, with $v(\emptyset) = 0$, that maps a *coalition* (*i.e.*, a set) of players to a real number representing the outcome or gain of a game (see, *e.g.*, [van den Brink et al., 2015]). A TU-game $(V, v)$ is *monotonic* if $v(S) \leq v(T)$ if $S \subseteq T \subseteq V$. Let us denote by $V^{-m}$ the set $V \setminus \{X_m\}$. The marginal contribution $MC_m^v(S)$ of player $X_m \in V$ for $S \subseteq V^{-m}$ is defined by $MC_m^v(S) = v(S \cup \{X_m\}) - v(S)$. A player $X_m \in V$ is called a *null player* if $MC_m^v(S) = 0$ for all $S \subseteq V^{-m}$. Two players $X_i$ and $X_j$ are said to be *symmetric* whenever $v(S \cup \{X_i\}) = v(S \cup \{X_j\})$ for all coalitions $S \subseteq V^{-i,j}$.

A *TU-value* $\varphi_v : V \to \mathbb{R}$ is a function that assigns to any player $X_m \in V$ and any function $v$ ($\in \mathbb{V}$ is omitted in the rest) a value, denoted $\varphi_v(X_m) \in \mathbb{R}$, also known as its *payoff*, reflecting its contribution in the game $(V, v)$. Several properties or axioms for TU-values have been defined in the literature that are expected to be satisfied in practical contexts (see [Besner, 2019] for a more exhaustive list):

**Efficiency:** For all $v$ , $\sum_{X_m \in V} \varphi_v(X_m) = v(V)$. *The TU-value divides the total gain (i.e., gain when all players are involved) among all players in an additive way.*

**Symmetry:** For all $v$, $\varphi_v(X_i) = \varphi_v(X_j)$ if players $X_i$ and $X_j$ are symmetric. *Two players of equal contributions in every game (i.e., with every coalition S) should get the same value.*

**Null player:** For all $v$ , $\varphi_v(X_m) = 0$ if $X_m$ is a null player. *A null player should get a zero payoff.*

**Strong monotonocity**[3]**:** For all $v, w$ and $X_m \in V$ such that $MC_m^v(S) \geq MC_m^w(S)$ for all $S \subseteq V^{-m}$, we have $\varphi_v(X_m) \geq \varphi_w(X_m)$. *If a player's marginal contributions are greater (or equal) in a game than in another in all coalitions, then its payoff in this game should not be lower than in the other.*

Specific forms of TU-value have been studied in the literature from the point of view of which axioms they satisfy and how uniquely they are defined by these axioms. As one of the most prominent results, it has been shown [Young, 1985] that the only TU-value that satisfies Efficiency, Symmetry, Null player[4], and Strong monotonicity is the *Shapley value* $\phi_v^{Sh}$ defined by [Shapley, 1953]:

$$\phi_v^{Sh}(X_m) = \sum_{S \subseteq V^{-m}} \frac{|S|!(p - |S| - 1)!}{p!} MC_m^v(S). \tag{1}$$

Other equivalent formulations of the Shapley value, as well as other axiomatisations of this value, have been proposed in the literature. Axiomatisations of other, typically more general, sets of TU-values are also available (see [Besner, 2019] for a recent and exhaustive discussion of this topic).

**Feature relevance.** In the feature selection literature, a common definition of the relevance of a feature is as follows [Kohavi et al., 1997]:

*A variable $X_m \in V$ is **relevant** to $Y$ (with respect to $V$) iff $\exists B \subset V : X_m \not\perp Y | B$. A variable $X_m$ is **irrelevant** if it is not relevant.* Relevant variables can be further divided into two categories according to their degree of relevance [Kohavi et al., 1997]: *A variable $X_m$ is **strongly** relevant*

---

[2]We use the same notations for the (set of) players as for (set of) input features, as the two will coincide later.

[3]In the ML literature, strong monotonicity is often called consistency [Lundberg and Lee, 2017].

[4]Actually, the Null player property is not required as it can be derived from strong monotonicity.

to $Y$ (with respect to $V$) iff $Y \not\perp X_m | V^{-m}$. A variable $X$ is **weakly** relevant if it is relevant but not strongly relevant. Strongly relevant variables thus convey information about the output that no other variable (or combination of variables) in $V$ conveys.

**Decision trees and forests.** Each interior node of a decision tree [Breiman et al., 1984] is labelled with a test based on some input and each leaf node is labelled with a value of the output. The tree is typically grown from a learning sample of size $N$ drawn from $P(V, Y)$ using a procedure that recursively partitions the samples at each node $t$ into two child nodes ($t_L$ and $t_R$). The test $s_t$ used to partition the samples at node $t$ is the one that maximises the mean decrease of some node impurity measure $i(\cdot)$ (*e.g.*, the Shannon entropy, the Gini index or the variance of $Y$): $\Delta i(s, t) = i(t) - \frac{p(t_L)}{p(t)} i(t_L) - \frac{p(t_R)}{p(t)} i(t_R)$, where $p(t_L)$ and $p(t_R)$ are the proportions of samples that fall in nodes $t_L$ and $t_R$ respectively. Single decision trees suffer from a high variance that is very efficiently reduced by building instead an ensemble of randomized trees and aggregating their predictions. Popular methods are Breiman [2001]'s Random Forests that build each tree from a different bootstrap sample with a local random selection of $K (\le p)$ variables at each node from which to identify the best split, and Geurts et al. [2006]'s Extra-Trees which skip bootstrapping and additionally randomly select the split values. Following Geurts et al. [2006], Louppe et al. [2013], ensemble of randomized trees grown with the value of the randomization parameters $K$ set to 1 will be called *Totally randomized trees*.

**Mean decrease impurity importance.** Given an ensemble of trees, several methods have been proposed to evaluate the (global) importance of variables for predicting the output [Breiman et al., 1984, Breiman, 2001]. This paper focuses on the Mean Decrease of Impurity (MDI) importance. Given a tree $T$, the MDI importance of a variable $X_m$ for predicting the output $Y$ is defined as :

$$\text{Imp}(X_m, T) = \sum_{t \in T : \nu(s_t) = X_m} p(t) \Delta i(s_t, t), \tag{2}$$

where the sum is over all interior nodes $t$ in $T$, $\nu(s_t)$ denotes the variable tested at node $t$, and $p(t)$ is the fraction of samples reaching node $t$. $\text{Imp}(X_m, T)$ is thus the (weighted) sum of impurity decreases over all nodes where $X_m$ is used to split. The MDI importance of $X_m$ derived from forests of $N_T$ trees is then the average of $\text{Imp}(X_m, T)$ over all trees:

$$\text{Imp}(X_m) = \frac{1}{N_T} \sum_T \text{Imp}(X_m, T). \tag{3}$$

While this measure was initially proposed as a heuristic, Louppe et al. [2013] characterise it theoretically under the following conditions: (1) all input variables and the output are categorical (not necessarily binary) (2) trees use so-called *exhaustive splits*[5], and (3) impurity is measured by Shannon entropy[6]. Later, we will refer to these conditions collectively as the *categorical setting*.

In the categorical setting, Louppe et al. [2013] (Thm. 1) show that for totally randomized trees (*i.e.*, $K = 1$) and in *asymptotic conditions* (*i.e.*, assuming $N_T \to \infty$ and a learning sample of infinite size), the MDI importance, denoted $\text{Imp}_\infty$, is given by:

$$\text{Imp}_\infty(X_m) = \sum_{k=0}^{p-1} \frac{1}{C_p^k} \frac{1}{p-k} \sum_{B \in \mathcal{P}_k(V^{-m})} I(Y; X_m | B), \tag{4}$$

where $\mathcal{P}_k(V^{-m})$ is the set of subsets of $V^{-m}$ of cardinality $k$, and $I(Y; X_m | B)$ is the conditional mutual information of $X_m$ and $Y$ given the variables in $B$. They also show that the sum of the MDI importances of all variables is equal to the mutual information between all input features and the output [Louppe et al., 2013, Thm. 2]:

$$\sum_{m=1}^{p} \text{Imp}_\infty(X_m) = I(Y; V). \tag{5}$$

A direct consequence of Equation 4 is that a variable $X_m$ is irrelevant iff $\text{Imp}_\infty(X_m) = 0$, which makes the MDI importance a sensible measure to identify relevant variables.

---

[5]Each node is split into $|\mathcal{X}_i|$ sub-trees, one for each of the $|\mathcal{X}_i|$ different values of the split variable $X_i$.

[6]A short introduction to information theory and the related notations used in the paper is provided in Appendix G.

## 3 *Global* MDI importances are Shapley values

In this section, we revisit the theoretical analysis of Louppe et al. [2013] and Sutera [2019] in the light of TU-games and TU-values, focusing on the categorical setting and asymptotic conditions adopted by these authors. We show in Section 3.1 that MDI importances computed from totally randomized trees can be interpreted as the Shapley value for a particular TU-game and we then discuss in Section 3.2 the case of non totally randomized trees (*i.e.*, $K > 1$).

### 3.1 Totally randomized trees

Let us consider a TU-game $(V, v)$, where $V$ is the set of variables and the coalition function $v$ is the mutual information $v(\cdot) = I(Y; \cdot)$. Since $v(\emptyset) = I(Y; \emptyset) = 0$, this is a valid TU-game. This TU-game is monotonic since we have $I(Y; T) = I(Y; S) + I(Y; T \setminus S | S) \geq I(Y; S)$ as soon as $S \subseteq T \subseteq V$ (using the chain rule and the positivity of the conditional mutual information). Marginal contributions for $v$ can be rewritten as:

$$MC_m^v(S) = v(S \cup \{X_m\}) - v(S) = I(Y; S \cup \{X_m\}) - I(Y; S) = I(Y; X_m | S), \qquad (6)$$

using the definition of (conditional) mutual information. A null player is thus defined as a variable $X_m$ such that $MC_m^v(S) = I(Y; X_m | S) = 0$ for all $S \subseteq V^{-m}$. This definition exactly coincides with the definition of an irrelevant variable (Section 2), since $I(Y; X_m | S) = 0$ is equivalent to $Y \perp X_m | S$. Two variables $X_i$ and $X_j$ are symmetric whenever $v(S \cup \{X_i\}) = v(S \cup \{X_j\})$ for all $S \subseteq V^{-i,j}$, which is equivalent to $I(Y; X_i | S) = I(Y; X_j | S)$ for all $S \subseteq V^{-i,j}$, *i.e.*, $X_i$ and $X_j$ bring the same information about $Y$ in all contexts $S$.

With this definition, the following theorem shows that MDI importance of totally randomized trees corresponds to the Shapley value for this TU-game:

**Theorem 1.** *(MDI are Shapley values) For all feature $X_m \in V$,*

$$Imp_\infty(X_m) = \phi_v^{Sh}(X_m), \qquad (7)$$

*where $\phi_v^{Sh}$ is the Shapley value with $v(S) = I(Y; S)$ ($\forall S \subseteq V$).*

The proof[7] of this theorem follows from a direct comparison of Equations 1 and 4.

Given this result, the four axioms that uniquely defines Shapley values are obviously satisfied. They translate into the following properties of the importances $Imp_\infty$:

**Efficiency**: $\sum_{m=1}^p Imp_\infty(X_m) = v(V) = I(Y; V)$, which states that MDI importances decompose, in an additive way, the mutual information $I(Y; X_1, \ldots, X_p)$. This results is identical to Equation 5.

**Symmetry**: If $X_i$ and $X_j$ are symmetric, then $Imp_\infty(X_i) = Imp_\infty(X_j)$. This property is easy to check knowing that $I(Y; S \cup \{X_i\}) = I(Y; S \cup \{X_j\})$ implies that $I(Y; X_i | S) = I(Y; X_j | S)$ and therefore swapping $X_i$ and $X_j$ in Equation 4 would keep all terms of the sum unchanged.

**Null player**: If $X_m$ is a null player, *i.e.*, an irrelevant variable, then $Imp_\infty(X_m) = 0$. Note that Louppe et al. [2013] actually showed a stronger result, stating in addition that $Imp_\infty(X_m) = 0$ *only if* $X_m$ is irrelevant to $Y$.

**Strong monotonicity**: Let us assume two outputs $Y_1$ and $Y_2$. Strong monotonicity says that if for all feature subsets $S \subseteq V^{-m}$: $I(Y_1; X_m | S) \geq I(Y_2; X_m | S)$, then we have $Imp_\infty^{Y_1}(X_m) \geq Imp_\infty^{Y_2}(X_m)$. This property states that if a variable brings more information about $Y_1$ than about $Y_2$ in all contexts $S$, it is more important to $Y_1$ than to $Y_2$.

The link between MDI importance and Shapley value shows that, in the finite setting, standard MDI importances from totally randomized trees compute an approximation of the Shapley values (for $I(Y; V)$), at least in the categorical setting[8]. MDI importance will be compared with other measures from the literature that explicitly seek to estimate the same quantities in Section 5.

---

[7]The proofs of all theorems are in Appendix A.

[8]For example, in the context of an ensemble of binary decision trees, MDI importance measure does not estimate the same quantities as in the categorical setting [Louppe, 2014, Sutera, 2019].

## 3.2 Non totally randomized trees

When $K > 1$, $K$ variables are randomly picked at each node and the best split, in terms of impurity reduction, among these $K$ variables is selected to actually split the node. Because several variables then compete for each split, some variables might be never (or less often) selected if there are other variables providing larger impurity decreases. These masking effects will impact the properties of the MDI importances. Let us denote by $\text{Imp}_\infty^K(X_m)$ the importance of $X_m$ derived from randomized trees built with a given value of $K$ in asymptotic conditions. When $K > 1$, $\text{Imp}_\infty^K(X_m)$ can no longer be decomposed as in Equation 4, as some $I(Y; X_m|S)$ terms will not be included in the sum or with a weight different from the one in Equation 4. Actually, although $\text{Imp}_\infty^K$ attributes a payoff to each variable in $V$, it can not be interpreted as a TU-value for the TU-game defined by $(V, v)$, with $v(\cdot) = I(Y; \cdot)$. Indeed, its computation requires to have access to conditional mutual information of the form $I(Y; X_m|S = s)$ for all coalition $S$ but also for all set of values $s$ of variables in $S$ and the latter can not be derived from the knowledge only of $I(Y; S)$ for all $S$.

The efficiency and null player conditions are however still satisfied by $\text{Imp}_\infty^K$:

**Efficiency**: Louppe et al. [2013] showed that $\sum_{m=1}^p \text{Imp}_\infty^K(X_m) = v(S) = I(Y; V)$ for all $K$ as soon as the trees are fully developed.

**Null player** If $X_m$ is a null player (*i.e.*, an irrelevant variable), then $\text{Imp}_\infty^K(X_m) = 0$ [Louppe et al., 2013]. Note that $\text{Imp}_\infty^K(X_m) = 0$ is however not anymore a sufficient condition for $X_m$ to be irrelevant. It can be shown however that a strongly relevant variable $X_m$ will always be such that $\text{Imp}_\infty^K(X_m) > 0$ whatever $K$ [Sutera et al., 2018].

Symmetry is not necessarily satisfied however, since $I(Y; X_i|S) = I(Y; X_j|S)$ for all $S \subseteq V^{-i,j}$ does not ensure that $I(Y; X_i|S = s) = I(Y; X_j|S = s)$ for all $s$, which would be required for the feature to be fully interchangeable when $K > 1$. Similarly, strong monotonicity is not satisfied either for $\text{Imp}_\infty^K$, as shown in Example 1 in Appendix B.

Note that as discussed in [Louppe et al., 2013], the loss of several properties when $K > 1$ should not preclude using $\text{Imp}_\infty^K$ as an importance measure in practice. In finite setting, using $K > 1$ may still be a sound strategy to guide the choice of the splitting variables towards the most impactful ones during tree construction and therefore result in more statistically efficient estimates.

## 4 *Local* MDI importances

So far, MDI importances are global, in that they assess the overall importance of each variable independently of any test instance. An important literature has emerged in the recent year that focuses on local importance measures that can assess the importance of a variable locally, *i.e.*, for a specific instance. In Section 4.1, we define a novel local MDI-based importance measure. We highlight the main properties of this measure and show in particular that it very naturally decomposes the standard global MDI measure. In Section 4.2, we show, in the categorical setting, that the asymptotic analysis of global MDI can be extended to the local MDI, which allows us, in Section 4.3 to show that local MDI importances are also Shapley values for a specific characteristic function in the case of totally randomized trees. Finally, in Section 4.4, we propose a local adaptation of the notion of feature relevance and link it with the local MDI measure.

### 4.1 Definition and properties

Let us denote by $\boldsymbol{x} = (x_1, \ldots, x_p)^T$ a given instance of the input variables, with $x_j$ the value of variable $X_j$. In what follows, we will further denote by $\boldsymbol{x}_S$ a given set of values for the variables in a subset $S \subseteq V$ (in particular, $\boldsymbol{x}_{\{X_j\}} = x_j$).

**Definition 1.** *(Local MDI) The local MDI importance $Imp(X_m, \boldsymbol{x})$ of a variable $X_m$ with respect to $Y$ for a given instance $\boldsymbol{x}$ is defined as follows*

$$Imp(X_m, \boldsymbol{x}) = \frac{1}{N_T} \sum_T \sum_{\substack{t \in T : \nu(s_t) = X_m \\ \wedge \boldsymbol{x} \in t}} i(t) - i(t_{x_m}) \tag{8}$$

*where the outer sum is over the $N_T$ trees of the ensemble, the inner sum is over all nodes that are traversed by $\boldsymbol{x}$ and where $X_m$ is used to split, $t_{x_m}$ is the successor of node $t$ followed by $\boldsymbol{x}$ in the tree (corresponding to $X_m = x_m$), and $i(.)$ is the impurity function.*

This general measure quantifies how important is feature $X_m$ to predict the output of the test example $\boldsymbol{x}$ represented by its input features. It collects all differences $i(t) - i(t_{x_m})$ along all paths traversed by example $\boldsymbol{x}$ in the ensemble. In practice, this can be implemented very efficiently at no additional cost with respect to the computation of a prediction, as soon as all impurities, computed at training time, are stored at tree nodes.

The intuition behind this measure is that a variable is important for a sample $\boldsymbol{x}$ if it leads to important reductions of impurity along the paths traversed by $\boldsymbol{x}$. Note that unlike global MDI, local MDI can be negative, as the impurity can increase from one node to one of its successors. A variable of negative importance for a given sample $\boldsymbol{x}$ is thus such that, in average over all paths traversed by $\boldsymbol{x}$, it actually increases the uncertainty about the output (because it helps for predicting the output of other instances).

A natural link between local and global MDI is given by the following result:

$$\text{Imp}(X_m) = \frac{1}{N} \sum_{i=1}^{N} \text{Imp}(X_m, \boldsymbol{x}^i), \tag{9}$$

where $\{(\boldsymbol{x}^1, y^1), \ldots, (\boldsymbol{x}^N, y^N)\}$ is the learning sample of $N$ examples that was used to grow the ensemble of trees. This result can be shown easily by combining the definitions in Equations 2 and 8 of global and local MDI respectively. Local MDI is thus a way to decompose the global MDI over all training examples.

## 4.2 Asymptotic analysis

In the categorical setting and asymptotic conditions, the decomposition in Equation 4 for totally randomized trees can be adapted to the local MDI measure, denoted $\text{Imp}_\infty(X_m, \boldsymbol{x})$.

**Theorem 2.** *(Asymptotic local MDI) The local MDI importance $Imp_\infty(X_m, \boldsymbol{x})$ of a variable $X_m$ with respect to $Y$ for a given sample $\boldsymbol{x}$ as computed with an infinite ensemble of fully developed totally randomized trees and an infinitely large training sample is*

$$Imp_\infty(X_m, \boldsymbol{x}) = \sum_{k=0}^{p-1} \frac{1}{C_p^k} \frac{1}{p-k} \sum_{B \in \mathcal{P}(V^{-m})} H(Y|B = \boldsymbol{x}_B) - H(Y|B = \boldsymbol{x}_B, X_m = x_m) \tag{10}$$

where $H(Y|\cdot)$ is the conditional entropy of $Y$. Similarly as in the finite setting, $\text{Imp}_\infty(X_m, \boldsymbol{x})$ can be negative, since the difference $H(Y|B = \boldsymbol{x}_B) - H(Y|B = \boldsymbol{x}_B, X_m = x_m)$ is not always positive. Example 2 in Appendix B illustrates one such a situation.

In asymptotic condition, the decomposition in Equation 9 furthermore becomes:

$$\text{Imp}_\infty(X_m) = \sum_{\boldsymbol{x} \in \mathcal{V}} P(V = \boldsymbol{x}) \text{Imp}_\infty(X_m, \boldsymbol{x}), \tag{11}$$

where the sum is over all possible input combinations. Combined with 5, this leads to the following double decomposition (over features and instances) of the information $I(V; Y)$:

$$I(V; Y) = \sum_{m=1}^{p} \sum_{\boldsymbol{x} \in \mathcal{V}} P(V = \boldsymbol{x}) \text{Imp}_\infty(X_m, \boldsymbol{x}). \tag{12}$$

## 4.3 Local MDI importances are Shapley values

Let us define a *local* characteristic function $v_{loc}(S; \boldsymbol{x}) = H(Y) - H(Y|S = \boldsymbol{x}_S)$, which measures the decrease in uncertainty (*i.e.*, the amount of information) about the output $Y$ when the variables in $S$ are known to be $\boldsymbol{x}_S$. This characteristic function is thus parameterized by $\boldsymbol{x}$. The proof of

Theorem 1 can be adapted to the decomposition in Equation 10 to show that local MDI importances of totally randomized trees in asymptotic conditions are Shapley values with respect to $v_{loc}(.; \boldsymbol{x})$:

$$\text{Imp}_\infty(X_m, \boldsymbol{x}) = \phi^{Sh}_{v_{loc}(.; \boldsymbol{x})}(X_m) \tag{13}$$

As a consequence, $\text{Imp}_\infty(X_m, \boldsymbol{x})$ satisfies the Shapley value properties at any point $\boldsymbol{x}$, *i.e.*:

**Efficiency**: $v_{loc}(V; \boldsymbol{x}) = H(Y) - H(Y|V = \boldsymbol{x}) = \sum_{m=1}^p \text{Imp}_\infty(X_m, \boldsymbol{x})$. This is in accordance with the decomposition in Equation 12, since $\sum_{\boldsymbol{x} \in \mathcal{V}} P(V = \boldsymbol{x}) v_{loc}(V; \boldsymbol{x}) = \sum_{x \in \mathcal{V}} P(V = \boldsymbol{x})(H(Y) - H(Y|V = \boldsymbol{x})) = I(V; Y)$.

**Symmetry**: If $X_i$ and $X_j$ are such that $H(Y) - H(Y|S = \boldsymbol{x}_S, X_i = x_i) = H(Y) - H(Y|S = \boldsymbol{x}_S, X_j = x_j)$ for every $S \subseteq V^{-i,j}$, then $\text{Imp}_\infty(X_i, \boldsymbol{x}) = \text{Imp}_\infty(X_j, \boldsymbol{x})$.

**Null player** If $X_i$ is such that $H(Y|S = \boldsymbol{x}_S, X_m = x_m) = H(Y|S = \boldsymbol{x}_S)$ for all $S \subseteq V^{-m}$, then $\text{Imp}_\infty(X_i, \boldsymbol{x}) = 0$.

**Strong monotonicity**: Assuming two outputs $Y_1$ and $Y_2$, if $H(Y_1|S = \boldsymbol{x}_S) - H(Y_1|S = \boldsymbol{x}_S, X_m = x_m) \geq H(Y_2|S = \boldsymbol{x}_S) - H(Y_2|S = \boldsymbol{x}_S, X_m = x_m)$ for all $S \subseteq V^{-m}$, then we have $\text{Imp}^{Y_1}(X_m, \boldsymbol{x}) \geq \text{Imp}^{Y_2}(X_m, \boldsymbol{x})$.

As in the case of global MDI, using non totally randomized trees ($K > 1$) will make local MDI to depart from the Shapley values, because of masking effects. Actually, local MDI importances will again not correspond to TU-values for $v_{loc}(\cdot; \boldsymbol{x})$, since they are not uniquely defined by $v_{loc}(\cdot; \boldsymbol{x})$. Indeed, tree splits along the paths traversed by $\boldsymbol{x}$ can not be determined from $v_{loc}(\cdot; \boldsymbol{x})$ only, as they depend on impurity reductions on other paths as well. However, the efficiency and null player properties will again remain valid, although symmetry and strong monotonicity are not guaranteed.

### 4.4 Local relevance

A major result regarding the global MDI importance in asymptotic conditions is its link with Kohavi et al. [1997]'s notion of feature relevance. A similar relationship can be established between local MDI importance measures and a new notion of local relevance (at the level of a samples) inspired from the null player property of Shapley values.

**Definition 2.** $X_m$ is ***locally irrelevant*** at $\boldsymbol{x}$ with respect to the output $Y$ iff $P(Y = y|X = x_m, B = \boldsymbol{x}_B) = P(Y = y|B = \boldsymbol{x}_B)$ for all $B \subseteq V^{-m}$ and all $y \in \mathcal{Y}$. It is ***locally relevant*** otherwise.

A variable is thus locally irrelevant at $\boldsymbol{x}$ if knowing its values never changes the probability of any output whatever the other variables that are known. Local relevance can be linked with global relevance through the following theorem.

**Theorem 3.** *A variable $X_m$ is irrelevant with respect to $Y$ if and only if it is locally irrelevant with respect to $Y$ for all $\boldsymbol{x}$ such that $P(V = \boldsymbol{x}) > 0$.*

In the categorical setting and asymptotic conditions, local relevance is linked to local MDI through the following theorem:

**Theorem 4.** *If a variable is locally irrelevant at $\boldsymbol{x}$ with respect to $Y$, then $\text{Imp}_\infty(X_m, \boldsymbol{x}) = 0$.*

Theorem 4 coincides exactly with the null player property of Section 4.3. Local MDI importance is thus a sensible score to identify locally irrelevant variables. Note that, unlike with global MDI, there might exist variables $X_m$ such that $\text{Imp}_\infty(X_m, \boldsymbol{x}) = 0$ despite $X_m$ being locally relevant at $\boldsymbol{x}$. However, a globally relevant variable $X_m$ will always receive a non zero $\text{Imp}_\infty(X_m, \boldsymbol{x})$ at some $\boldsymbol{x}$.

## 5 Discussion and related works

In the literature, Shapley values have been mostly used to decompose model predictions $\hat{f}(\boldsymbol{x})$ at any $\boldsymbol{x}$ into a sum of terms that represent the (local) contribution of each variable to the prediction [Strumbelj and Kononenko, 2010, Lundberg and Lee, 2017]. The characteristic function $v_{\hat{f}}$ considered by these methods is $v(S) = \hat{f}_S(\boldsymbol{x}_S) - \hat{f}_\emptyset(\boldsymbol{x}_\emptyset)$, where $\hat{f}_S(\boldsymbol{x}_S)$ is the model to be explained restricted to the variables in $S$ and $\hat{f}_\emptyset(\boldsymbol{x}_\emptyset)$ is often set to $E[\hat{f}(X)]$. Typically, $\hat{f}_S(\boldsymbol{x}_S)$ is defined as $\mathbb{E}[\hat{f}(X)|X_S = \boldsymbol{x}_S]$, where the expectation is over the conditional $p(X_{\bar{S}}|X_S = \boldsymbol{x}_S)$ or the marginal (a.k.a. interventional)

$p(X_{\bar{S}})$ distribution. These methods are mostly model agnostic, *i.e.*, they can handle any machine learning model, considered as a black-box, although the estimation of restricted models and the computation of the Shapley values (Equation 1) can be very challenging in general.

Among this literature, Lundberg et al. [2020] have proposed TreeSHAP, a framework to efficiently compute Shapley values when $\hat{f}$ are trees or sum of trees. One of the only alternative local importance measures for trees is Saabas' heuristic method (implemented in [Saabas, 2014]). Saabas' method measures local variable importances for a sample $\boldsymbol{x}$ by collecting the changes in the (expected) model prediction due to each variable value along the tree branches followed by $\boldsymbol{x}$. Like TreeSHAP, Saabas' importances sum to the model prediction at $\boldsymbol{x}$. They are much faster to compute but do not satisfy all properties of Shapley values, in particular strong monotonicity.

One main difference between local MDI and TreeSHAP/Saabas as studied in Lundberg et al. [2020] is that local MDI decomposes entropy (or more generally impurity) reductions ($v_{loc}(V; \boldsymbol{x})$), while TreeSHAP/Saabas decompose model predictions[9] ($v_{\hat{f}}$). As a consequence, local MDI scores are independent of output normalisation or scaling and do not require to choose a specific class probability score to be used as $\hat{f}(\boldsymbol{x})$ to be decomposed. This also allows to connect local and global MDI in a natural way, and gives a probabilistic interpretation to the null player property in terms of variable irrelevance. Algorithmically, local MDI uses the exact same collecting procedure along tree paths as Saabas' measure, replacing output differences with impurity reductions. Similarly as Saabas, local MDI results in a much more efficient and simpler estimation scheme than TreeSHAP but it looses some properties of Shapley values in the general case. Section 4.3 however shows that these properties are retrieved in the case of totally randomized trees, at least asymptotically. This also applies to Saabas' measure (as sketched in Theorem 1 in the supplement of Lundberg et al. [2020]). Although TreeSHAP is guaranteed to ensure strong monotonicity asymptotically, the relevance of its scores is still tied to the quality of the tree-based model that it explains. For example, using TreeSHAP with Random forests with $K > 1$ or pruned trees will also potentially lead to biases in the importance scores (*e.g.*, due to masking effects) with respect to what would be obtained if $\hat{f}$ was the Bayes classifier. We believe this is a similar trade-off as the one met in local MDI with respect to $K$.

Our results also highlight a link between global MDI and SAGE [Covert et al., 2020], a purely model-agnostic method for global importance computation. SAGE estimates Shapley values for $v_\ell(S) = \mathbb{E}[\ell(\hat{f}_\emptyset(X_\emptyset), Y)] - \mathbb{E}[\ell(\hat{f}_S(X_S), Y)]$ where $\ell$ is a loss function and expectations are taken over $p(V, X)$. Covert et al. [2020] have shown that when $\ell$ is cross-entropy, $\hat{f}$ is the Bayes classifier, and restricted models are estimated through the conditional distributions, then the population version of SAGE is strictly identical to Equation 4. Interestingly, both methods arrive to this population formulation through very different algorithms. SAGE explicitly estimates Shapley values, while global MDI are obtained by collecting impurity reductions at tree nodes in a random forest. Global MDI departs from 4 and Shapley values when $K > 1$. On the other hand, being model agnostic, like TreeSHAP, SAGE is tied to the quality of the model it explains. Given the difficulty of sampling from the conditional distribution, its implementation also samples from the marginal distribution instead, which makes it depart from Shapley values and affects its convergence to 4. In practice, we will show in the next section that both methods produce very similar results, when used with Random forests (but with a strong advantage to global MDI in terms of computing times).

Overall, we believe our analysis of local and global MDI sheds some new lights on these measures. Although they were not designed as such, these methods can indeed be interpreted as procedures to sample variable subsets and compute mutual information such that they provide estimates of Shapley values for a very natural characteristic function based on mutual information. Although they are tightly linked algorithmically with Random forests, they actually highlight general properties of the original data distribution independently of these models. This makes them very different from model-agnostic methods that explain pre-existing models, furthermore regardless of the data distribution when restricted models are estimated from marginal distributions.

---

[9]Although TreeSHAP authors advocate the decomposition of model predictions as the way to go, a variant of TreeSHAP [Lundberg et al., 2020] can also decompose model loss by enforcing feature independence, at a higher computational cost however than the local MDI measure proposed here.

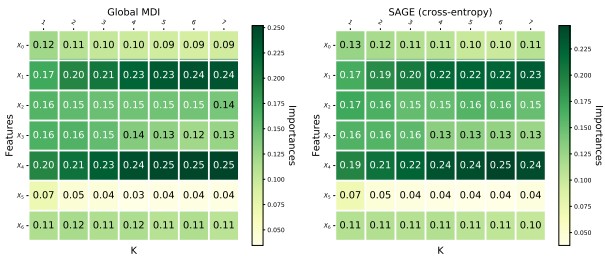
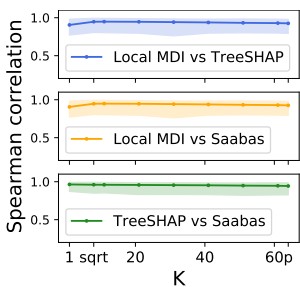

Figure 1: Normalized importance scores derived from an ensemble of totally and non-totally randomized Extra-Trees (with $K = 1, \dots, p$) for the global MDI importance measure (left) and SAGE (right).

Figure 2: Mean correlation (over all samples) w.r.t. increasing $K$ between absolute importance scores for `digits` ([min,max] is shaded).

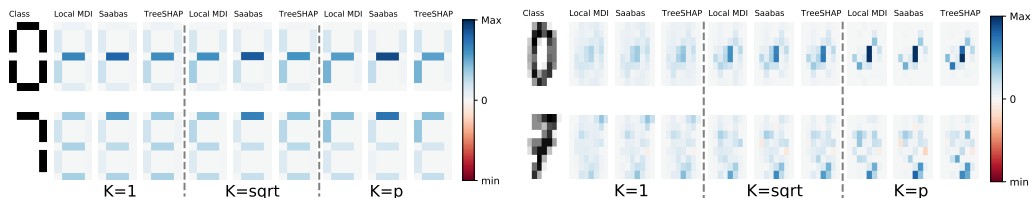

Figure 3: Local importances derived by local measures from a forest of 1000 Extra-Trees with $K \in \{1, \sqrt{p}, p\}$) for `led` (left) and `digits` (right). Results for all samples can be found in Appendix E.

## 6    Illustrations

Here, global and local MDI are illustrated and compared against SAGE, Saabas, and TreeSHAP on two classification problems. The `led` problem [Breiman et al., 1984] consists of an output $Y \in \{0, \dots, 9\}$ with equal probability and seven binary variables representing each a segment of a seven-segment display and whose values are determined unequivocally from $Y$. The `digits` problem consists of an output $Y \in \{0, \dots, 9\}$ and 64 integer inputs corresponding to the pixels of a $8 \times 8$ grayscale image of a digit. Additional experiments on other datasets are reported in Appendix E. In all experiments, importance scores are computed either using Scikit-Learn [Pedregosa et al., 2011] or method authors' original code (data and code are open-source, see details in Appendix C).

**Global importances** are shown in Figure 1 on the `led` dataset for an ensemble of 1000 Extra-Trees [Geurts et al., 2006] and several values of $K$, with global MDI and SAGE (using the cross-entropy loss). Importances are normalized such that the sum of (absolute values of) the scores is equal to one. With $K = 1$, both approaches clearly yield very similar scores, as expected from the discussion in Section 5. Both methods remain very close when $K$ is increased. Regardless of the importance measure, one can notice the masking effect that favors $X_1$ and $X_4$ at the expense of the other variables when $K$ increases. The same effect appears when SAGE uses another loss function and for the global importance measure derived from TreeSHAP (see Appendix D).

**Local importances** are then computed for both classification problems from an ensemble of 1000 Extra-Trees and with the three local importance measures. Scores from Saabas and TreeSHAP are obtained from the additive decomposition of the conditional probability of the predicted class (which might not be the true class). Local feature importances for two samples are reported in Figure 3 by coloring either the corresponding segment (left) or pixel (right) for three values of $K$. The three local importance measures yield very similar importance scores, suggesting that they provide matching explanations of model predictions. This similarity is further quantified by measuring the correlation between the (absolute value of) local importance scores of all pairs of methods for several $K$ (Figure 2 for `digits`, Appendix E for `led`). The mean correlation over all samples remains close to 1 for all pairs, although the variation (depicted by the shaded area delimiting the range between minimal and maximal correlations) is impacted by the value of $K$. As expected, Saabas and TreeSHAP, which both decompose model predictions, are closer to each other than to Local MDI.

# 7 Conclusion

MDI importances have been used extensively as a way of measuring, globally, the respective contribution of variables. In this paper, we showed that global MDI importances derived from totally randomized trees are actually Shapley values that decompose the mutual information $I(V; Y)$ when the impurity measure is the Shannon entropy. We then proposed a local MDI importance that very naturally decomposes global MDI over the training examples. We showed that local MDI importances are also Shapley values with respect to conditional entropy reductions and that they are consistent with respect to a novel local relevance notion. We compared MDI importances conceptually with other recent local and global feature importance scores inspired from Shapley values and showed empirically that all these methods are very close, while both global and local MDI importances do not require any extra computation with respect to the tree construction. Overall, local and global MDI measures provide a natural and efficient way of explaining properties of the data distribution.

While the main results of this paper assume categorical variables and Shannon entropy as impurity measure, they can be extended to other impurity measures and to regression (see Appendix F). As future works, we would like to better characterize these measures in non-asymptotic conditions and outside of the purely categorical setting. More experiments should be also carried out to better highlight differences, practically, between the MDI family and other methods such as TreeSHAP and SAGE that more explicitly approximate Shapley values. Finally, the link with Shapley values and TU-games in general could be further investigated to propose other extensions of MDI measures (for example to highlight variables interactions as in [Lundberg et al., 2020]).

## Acknowledgments

Antonio Sutera is supported via the Energy Transition Funds project EPOC 2030-2050 organized by the FPS economy, S.M.E.s, Self-employed and Energy. This work was partially supported by Service Public de Wallonie Recherche under grant n° 2010235 – ARIAC by DIGITALWALLONIA4.AI.

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
