# From global to local MDI variable importances for random forests and when they are Shapley values
## *Supplementary materials*

**Antonio Sutera,**[*] **Gilles Louppe, Van Anh Huynh-Thu, Louis Wehenkel, Pierre Geurts**
Dept. of EE & CS, University of Liège, Belgium
{a.sutera,g.louppe,vahuynh,l.wehenkel,p.geurts}@uliege.be

## A  Proofs

### A.1  Proof of Theorem 1

**Theorem 1.** *(MDI are Shapley values) For all feature $X_m \in V$,*

$$Imp_\infty(X_m) = \phi_v^{Sh}(X_m), \tag{8}$$

*where $\phi_v^{Sh}(X_m)$ is the Shapley value of $X_m$ with respect to the characteristic function $v(S) = I(Y; S)$ (with $S \subseteq V$).*

*Proof.* Let us first note that

$$
\begin{aligned}
v(S \cup \{X_m\}) - v(S) &= I(Y; S, X_m) - I(Y; S) \\
&= H(Y) - H(Y|S, X_m) - H(Y) + H(Y|S) \\
&= I(X_m; Y|S)
\end{aligned} \tag{1}
$$

Replacing the characteristic functions as defined in this equation in Equation 1 of the main paper, Shapley values can thus be defined as:

$$\phi_v(X_m) = \sum_{S \subseteq V^{-m}} \frac{|S|!(p - |S| - 1)!}{p!} I(X_m; Y|S) \tag{2}$$

The sum can be reorganized according to the size of the subsets $S$ from $V^{-m}$:

$$\phi_v(X_m) = \sum_{k=0}^{p-1} \frac{k!(p - k - 1)!}{p!} \sum_{S \subseteq \mathcal{P}_k(V^{-m})} I(X_m; Y|S) \tag{3}$$

which is strictly equivalent to $Imp_\infty(X_m)$ given that:

$$
\begin{aligned}
\frac{k!(p - k - 1)!}{p!} &= \frac{(p - k)!k!}{p!} \frac{1}{p - k} \\
&= \frac{1}{\dfrac{p!}{(p - k)!k!}} \frac{1}{p - k} \\
&= \frac{1}{C_p^k} \frac{1}{p - k}
\end{aligned} \tag{4}
$$

$\square$

---

[*]Corresponding author. Email address: sutera.antonio@gmail.com

35th Conference on Neural Information Processing Systems (NeurIPS 2021).

## A.2 Proof of Theorem 2

**Theorem 2.** *(Asymptotic local MDI) The local MDI importance $Imp_\infty(X_m, \boldsymbol{x})$ of a variable $X_m$ with respect to $Y$ for a given sample $\boldsymbol{x}$ as computed with an infinite ensemble of fully developed totally randomized trees and an infinitely large training sample is*

$$Imp_\infty(X_m, \boldsymbol{x}) = \sum_{k=0}^{p-1} \frac{1}{C_p^k} \frac{1}{p-k} \sum_{B \in \mathcal{P}(V^{-m})} H(Y|B = \boldsymbol{x}_B) - H(Y|B = \boldsymbol{x}_B, X_m = x_m) \quad (11)$$

*Proof.* Let $B(t) = (X_{i_1}, \ldots, X_{i_k})$ be the subset of $k$ variables tested in the branch from the root node to the parent of $t$ and $b(t)$ be the vector of values of these variables. As the number of training samples grows to infinity, the probability that a sample reaches node $t$ is $P(B(t) = b(t))$ (according to $P(X_1, \ldots, X_p, Y)$). As the number $N_T$ of totally randomized trees also grows to infinity, the local importance of variable $X_m$ for sample $\boldsymbol{x}^i$ can then be written:

$$Imp(X_m, \boldsymbol{x}) = \sum_{B \subseteq V^{-m}} \beta \left( H(Y|B = \boldsymbol{x}_b) - H(Y|B = \boldsymbol{x}_B, X_m = x_m) \right) \quad (5)$$

where $\beta$ is probability that a node $t$ (at depth $k$) in a totally randomized tree tests the variable $X_m$ and is such that $B(t) = B$ and $b(t) = \boldsymbol{x}_B$. $\beta$ is given by [Louppe et al., 2013] as being equal to $\frac{1}{C_p^k} \frac{1}{p-k}$ and remains valid because it only depends on the size $k$ of $B$ and on the number $p$ of variables. Notice already the similarity with the intermediate formulation in the proof of Theorem 1 from [Louppe et al., 2013] where Equation 5 reduces the inner sum to a single term, the one corresponding to the given $b = \boldsymbol{x}_B$. Rewritting Equation 5 in order to group subsets $B$ according to their sizes, we have

$$Imp_\infty(X_m, \boldsymbol{x}) = \sum_{k=0}^{p-1} \frac{1}{C_p^k} \frac{1}{p-k} \sum_{B \in \mathcal{P}(V^{-m})} H(Y|B = \boldsymbol{x}_B) - H(Y|B = \boldsymbol{x}_B, X_m = \boldsymbol{x}_m) \quad (6)$$

where the inner sum is over the set of subsets of $V^{-m}$ of cardinality $k$ (*i.e.*, the different paths of length $k$ leading to a test on $X_m$), and which completes the proof. $\square$

## A.3 Proof of Theorem 3

**Theorem 3.** *(Equivalence of irrelevance) A variable $X_m$ is irrelevant with respect to $Y$ if and only if it is locally irrelevant with respect to $Y$ for all $\boldsymbol{x}$ such that $P(V = \boldsymbol{x}) > 0$.*

*Proof.* This proof directly stems from the following intuitive observation: the irrelevance property considers all $\boldsymbol{x}$ while the local irrelevance one only considers one $\boldsymbol{x}$. If local irrelevance is satisfied for all $\boldsymbol{x}$, then irrelevance is satisfied. The other way around is trivial. Thus the irrelevance property is equivalent to the set of local irrelevance properties corresponding to each $\boldsymbol{x}$. Mathematically, we can also prove this equivalence as follows. By definition of irrelevance, $X_m \perp Y|B$ for all $B \subseteq V^{-m}$, where $X_m \perp Y|B$ is the conditional independence and is equivalent (in the case of discrete variables and assuming $P(B = b) > 0$) to saying that $P(X_m = x_m, Y = y|B = b) = P(X_m = x_m|B = b)P(Y = y|B = b)$ for all $y, x_m$. It comes that $P(Y = y|X_m = x_m, B = b) = P(Y = y|B = b)$ and so both notions are equivalent if the local definition is valid for all $y$ and $x_m$.

Note that this proof can be extended to continuous variables by changing probabilities $P(X = x)$ to $P(X \le x)$. $\square$

## A.4 Proof of Theorem 4

**Theorem 4.** *If a variable is locally irrelevant at $\boldsymbol{x}$ with respect to $Y$, then $Imp_\infty(X_m, \boldsymbol{x}) = 0$.*

*Proof.* The proof stems from the definition of the local irrelevance. By definition, if $X_m$ is locally irrelevant at $\boldsymbol{x}$ with respect to the output $Y$, then $P(Y = y|X_m = x_m, B = \boldsymbol{x}_B) = P(Y = y|B = $

$\boldsymbol{x}_B$) for all $B \subseteq V^{-m}$ and all $y \in \mathcal{Y}$. Consequently,

$$
\begin{aligned}
H(Y|B = \boldsymbol{x}_B, X_m = x_m) &= -\sum_{y \in \mathcal{Y}} P(y|\boldsymbol{x}_B, x_m) \log P(y|\boldsymbol{x}_B, x_m) \\
&= -\sum_{y \in \mathcal{Y}} P(y|\boldsymbol{x}_B) \log P(y|\boldsymbol{x}_B) \\
&= H(Y|B = \boldsymbol{x}_B).
\end{aligned}
$$

If $X_m$ is locally irrelevant at $\boldsymbol{x}$, each collected term is therefore equal to zero, leading to $Imp_\infty(X_m, \boldsymbol{x}) = 0$.

$\square$

## B Examples

**Example 1.** *Let us consider the two binary classification problems, with outputs $Y_1$ and $Y_2$ and two binary inputs $X_1$ and $X_2$, described in Table 1. One can compute the following conditional mutual information terms:*

$$
\begin{aligned}
I(Y_1; X_1) &= 0.091 \quad (\geq) \quad I(Y_2; X_1) = 0.002, \\
I(Y_1; X_1|X_2) &= 0.269 \quad (\geq) \quad I(Y_2; X_1|X_2) = 0.243, \\
I(Y_1; X_2) &= 0.002 \quad (\leq) \quad I(Y_2; X_2) = 0.016, \\
I(Y_1; X_2|X_1) &= 0.180 \quad (\leq) \quad I(Y_2; X_2|X_1) = 0.258.
\end{aligned}
$$

*If $K = 2$, the forest reduces to a single tree both for $Y_1$ and $Y_2$. For $Y_1$, this tree first splits on $X_1$ and then on $X_2$, resulting in the following importances:*

$$
\begin{aligned}
Imp_\infty^{K=2, Y_1}(X_1) &= I(Y_1; X_1) = 0.091, \\
Imp_\infty^{K=2, Y_1}(X_2) &= I(Y_1; X_2|X_1) = 0.180.
\end{aligned}
$$

*For $Y_2$, the tree first splits on $X_2$ and then on $X_1$, resulting in the following importances:*

$$
\begin{aligned}
Imp_\infty^{K=2, Y_2}(X_1) &= I(Y_2; X_1|X_2) = 0.243, \\
Imp_\infty^{K=2, Y_2}(X_2) &= I(Y_2; X_2) = 0.016.
\end{aligned}
$$

*One can see that the strong monotonicity property is violated both for $X_1$ and $X_2$. For example, although $X_1$ brings more information about $Y_1$ than about $Y_2$ in all contexts, it is more important to $Y_2$ than $Y_1$. This is due to the fact that in the tree for $Y_2$, $X_1$ appears at the second level of the tree and it thus receive more credit than in the tree for $Y_1$ where it appears at the top node.*

Table 1: Definition of two outputs for which the strong monotonicity constraint is not satisfied, neither for $X_1$, nor for $X_2$. All input combinations are assumed to be equiprobable.

| $X_1$ | $X_2$ | $P(Y_1 = 1|X_1, X_2)$ | $P(Y_2 = 1|X_1, X_2)$ |
|---|---|---|---|
| 0 | 0 | 0.1 | 0.1 |
| 0 | 1 | 0.5 | 0.8 |
| 1 | 0 | 0.9 | 0.7 |
| 1 | 1 | 0.4 | 0.3 |

**Example 2.** *Let us consider a binary classification problem with a single binary input $X_1$ and assume that $P(Y = 0) > P(Y = 1)$ and $P(Y = 0|X_1 = 0) = P(Y = 1|X_1 = 0)$. In this case, $Imp_\infty(X_1, 0) = H(Y) - H(Y|X_1 = 0)$, which is negative. Table 2 gives a numerical example of this.*

Table 2: Definition of $X_1$ and $Y$ such that $Imp_\infty(X_1, 0) < 0$. Here, $Imp_\infty(X_1, 0) = 0.81 - 1 = -0.19$. All input combinations are assumed to be equiprobable.

| $X_1$ | $P(Y=0\|X_1)$ | $P(Y=1\|X_1)$ | | | $P(Y)$ | |
|---|---|---|---|---|---|---|
| 0 | 0.5 | 0.5 | $H(Y\|X_1=0)=1$ | 0 | 0.75 | $H(Y)=0.81$ |
| 1 | 1.0 | 0.0 | $H(Y\|X_1=1)=0$ | 1 | 0.25 | |

## C   Code and data

Tree-based models are computed using Scikit-Learn [Pedregosa et al., 2011] (BSD-3-Clause License) and other importance measures are computed using the corresponding latest available source code: Saabas (BSD-3-Clause License) from TREEINTERPRETER '*v0.1.0*', TreeSHAP from SHAP 'v0.38.2' (MIT License), and SAGE from its Github repository (MIT License). As there is no versioning of the SAGE package, the code used was lastly downloaded on the $6^{th}$ of February 2021. Global MDI is computed using Scikit-Learn. Source code (BSD-3-Clause License) to compute local MDI is available (open-source) at https://github.com/asutera/Local-MDI-importance.

## D   Supplementary results for global importance measures

Figure 1 reports normalized importance scores derived from an ensemble of trees with increasing $K$ (*i.e.*, the randomization parameter) on the Led dataset for SAGE with the mean squared loss (mse) as loss function, and the mean of the absolute value of TreeSHAP (for the predicted class) with both available parameters for feature perturbations. It appears that all three methods reflect the impact of $K$ on the measured importance scores similarly as the global MDI and SAGE using the cross-entropy loss function (Figure 1).

## E   Supplementary results for local importance measures

This section presents additional results and experiments that compare local importance measures.

### E.1   Local importances for led and digits

Figure 3 shows the correlation between the (absolute value) of local importance scores of all pairs of methods for several values of the randomization parameter $K$. Figures 2 reports the local importances for three values of $K$ for led (left) and digits (right), showing more samples (all samples of led and one of each class for digits) than Figure 3.

### E.2   Local importances on other datasets

In addition to the led and digits datasets already used in Section 6, we consider here a few additional classification datasets. led (sampled) is a variant of the foretold problem where the learning set is made of (200) samples randomly drawn from the data distribution. The remaining datasets have been chosen from Scikit-learn datasets to cover mixed settings and differ by their dimensionality and their feature types (both discrete and continuous). Table 3 summarizes their characteristics.

Table 3: Datasets in classification

| Name | Nb of samples | Nb of features | Feature types |
|---|---|---|---|
| Led | 10 | 7 | Binary |
| Led (sampled) | 200 | 7 | Binary |
| Digits | 1797 | 64 | Integer |
| Iris | 150 | 4 | Real |
| Wine | 178 | 13 | Integer, Real |
| Breast cancer | 569 | 30 | Real |

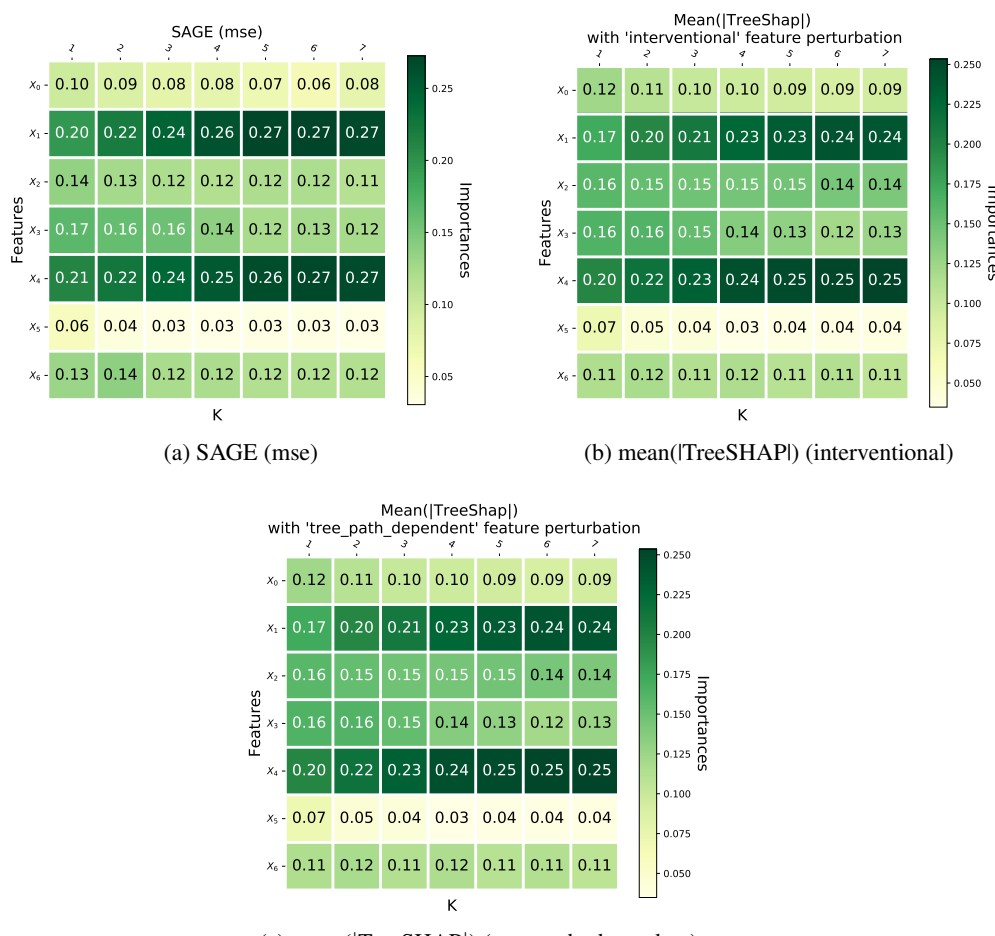

(a) SAGE (mse)

(b) mean(|TreeSHAP|) (interventional)

(c) mean(|TreeSHAP|) (tree_path_dependent)

Figure 1: Normalized importance scores derived from an ensemble of totally and non-totally randomized Extra-Trees (with $K = 1, \ldots, p$) for the global MDI importance measure and SAGE.

For every dataset, local feature importances are derived from a forest of 1000 totally randomized Extra-Trees ($K = 1$). As in Section 6, Saabas and TreeSHAP are computed with respect to the predicted class.

For each sample, Pearson and Spearman correlations are computed between the local MDI and Saabas importances and between the local MDI and treeSHAP importances.

Figure 4 shows the correlations across all samples that are reordered by increasing correlation values. Table 4 summarizes these results.

It can be observed that a large fraction of samples are associated to correlations close to 1 suggesting that most of importance vectors are similar. More than $90\%$ importance vectors are similar (correlation $\geq 0.75$) in four datasets (out of six) and more than $80\%$ in two datasets. Note that this also shows that a fraction of samples (roughly $1\%$ to $10\%$ depending on the dataset) are associated with uncorrelated importance scores. This may come from the importance scores with respect to the predicted class (used in TreeSHAP and Saabas) that differs from the importance scores over all classes (local MDI). More experiments should be carried out to better understand the origin of these differences.

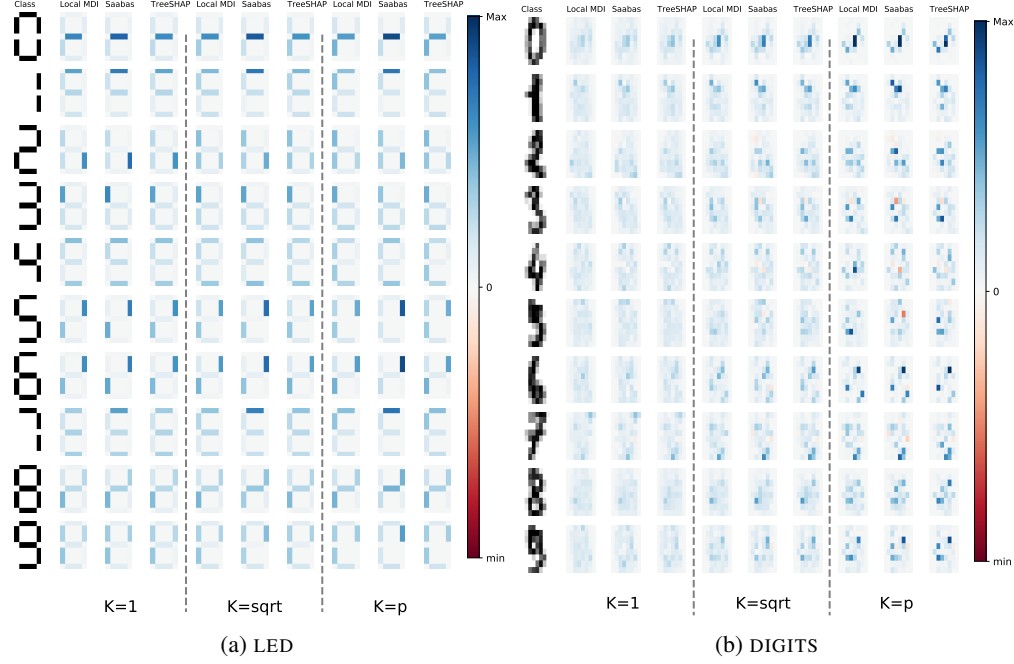

| (a) LED | (b) DIGITS |

Figure 2: Local importances derived by local measures from a forest of 1000 Extra-Trees with $K \in \{1, \sqrt{p}, p\}$) for led (left) and digits (right).

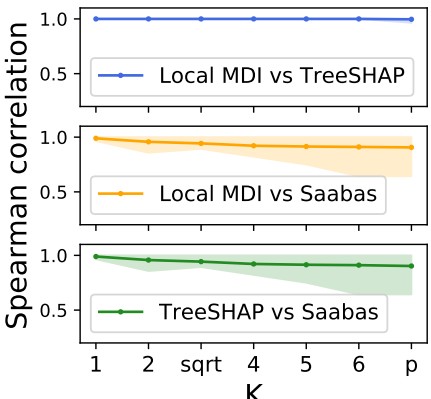

Figure 3: Mean correlation (over the samples) w.r.t. increasing $K$ for between absolute importance scores for led ([min,max] is shaded).

# F    Generalization to other impurity measures and to regression

We have considered in most of our developments in the paper a categorical output $Y$ (*i.e.*, a classification problem) and the use of Shannon entropy as impurity measure. Louppe et al. [2013] show that Equation 4 and the link between the irrelevance of $X_m$ and $Imp_\infty(X_m)$ remain valid for other impurity measures in classification, such as the Gini index, and can be extended to regression problems using variance as the impurity measure. Similarly, the local MDI measure can be extended to other impurity measures and thus in particular also to regression problems (*i.e.*, a numerical output $Y$). Definition 8 is indeed generic and valid whatever the function $i(Y|t) \geq 0$ that measures the impurity of the output $Y$ at a tree node $t$. The link between local and global MDI as expressed in Equation 9 (main paper) is also generic. Asymptotic results in Sections 4.2 and 4.3, i.e., the decomposition in Theorem 2 and the link with Shapley value, remain valid with entropy $H$ replaced by the population version of the choosen impurity measure, denoted $I_\infty$ in what follows. Results in Section 4.4 requires

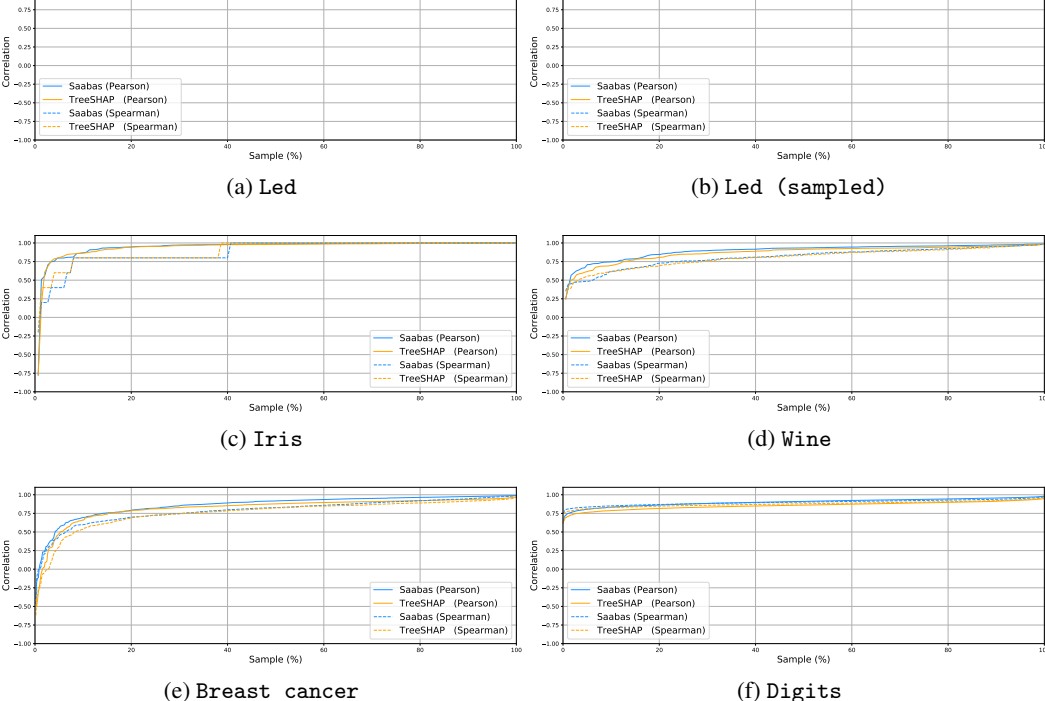

(a) `Led`

(b) `Led (sampled)`

(c) `Iris`

(d) `Wine`

(e) `Breast cancer`

(f) `Digits`

Figure 4: Correlations between feature importance vectors across the samples ordered by increasing values.

to redefine the notion of local irrelevance from the impurity function directly. If one defines a variable $X_m$ as *locally irrelevant* at $\boldsymbol{x}$ w.r.t. $Y$ iff $I_\infty(Y|X_m = \boldsymbol{x}_m, B = \boldsymbol{x}_B) = I_\infty(Y|B = \boldsymbol{x}_B)$ for all $B \subseteq V^{-m}$ [2], then Theorem 4 still applies.

As an illustration, if one considers regression using the empirical variance as the impurity measure:

$$i(Y|t) = \frac{1}{N_t} \sum_{i \in t} (y_i - \frac{1}{N_t} \sum_{i \in t} y_i)^2,$$

where $N_t$ is the number of instances in node $t$ and $y_i$ are their output values, then $I_\infty$ is the (conditional) population variance and the decomposition in Equation 10 (main paper) becomes:

$$Imp_\infty(X_m, \boldsymbol{x}) = \sum_{k=0}^{p-1} \frac{1}{C_p^k} \frac{1}{p-k} \sum_{B \in \mathcal{P}(V^{-m})} \text{Var}(Y|B = \boldsymbol{x}_B) - \text{Var}(Y|B = \boldsymbol{x}_B, X_m = x_m), \quad (7)$$

where $\text{Var}(Y|B = \boldsymbol{x}_B)$ is the conditional variance of the output:

$$\text{Var}(Y|B = \boldsymbol{x}_B) = E_{Y|B=\boldsymbol{x}_B}\{(Y - E_{Y|B=\boldsymbol{x}_B}\{Y\})^2\}. \quad (8)$$

Global and local MDI importances of totally randomized trees are in this case Shapley values respectively with respect to the following characteristic functions:

$$v(S) = \text{Var}(Y) - E_S\{\text{Var}(Y|S)\} \quad (9)$$
$$v_{loc}(S; \boldsymbol{x}) = \text{Var}(Y) - \text{Var}(Y|S = \boldsymbol{x}_S), \quad (10)$$

and the decomposition in Equation 12 therefore becomes:

$$\text{Var}(Y) - E_S\{\text{Var}(Y|X)\} = \sum_{m=1}^{p} \sum_{\boldsymbol{x} \in \mathcal{V}} P(V = \boldsymbol{x}) Imp_\infty(X_m, \boldsymbol{x}). \quad (11)$$

Finally, locally irrelevant variables (null players) are such that $\text{Var}(Y|X_m = \boldsymbol{x}_m, B = \boldsymbol{x}_B) = \text{Var}(Y|B = \boldsymbol{x}_B)$ for all $B \in V^{-m}$.

---

[2]This definition matches Definition 2 when $I_\infty$ is the entropy $H$.

Table 4: Summary of results on classification datasets.

| Saabas | | | | |
|---|---|---|---|---|
| Dataset | Correlation | Avg. corr. (±std) | Fraction of samples | |
| | | | w/ correlation $\geq 0.9$ | w/ correlation $\geq 0.75$ |
| Wine | Pearson | 0.906 (±0.101) | 72.47% | 91.57% |
| | Spearman | 0.843 (±0.128) | 44.38% | 80.90% |
| Iris | Pearson | 0.949 (±0.156) | 89.33% | 95.33% |
| | Spearman | 0.892 (±0.231) | 67.33% | 91.33% |
| Breast cancer | Pearson | 0.899 (±0.220) | 79.96% | 91.04% |
| | Spearman | 0.857 (±0.255) | 68.19% | 84.71% |
| Led | Pearson | 0.980 (±0.025) | 100.00% | 100.00% |
| | Spearman | 0.989 (±0.016) | 100.00% | 100.00% |
| Led (sampled) | Pearson | 0.970 (±0.034) | 100.00% | 87.00% |
| | Spearman | 0.978 (±0.017) | 100.00% | 100.00% |
| Digits | Pearson | 0.915 (±0.045) | 69.84% | 99.44% |
| | Spearman | 0.899 (±0.045) | 55.65% | 99.50% |

| TreeSHAP | | | | |
|---|---|---|---|---|
| Dataset | Correlation | Avg. corr. (±std) | Fraction of samples | |
| | | | w/ correlation $\geq 0.9$ | w/ correlation $\geq 0.75$ |
| Wine | Pearson | 0.900 (±0.104) | 70.22% | 90.45% |
| | Spearman | 0.852 (±0.121) | 47.75% | 82.02% |
| Iris | Pearson | 0.947 (±0.150) | 88.00% | 96.67% |
| | Spearman | 0.881 (±0.186) | 56.67% | 92.00% |
| Breast cancer | Pearson | 0.888 (±0.223) | 79.26% | 90.33% |
| | Spearman | 0.841 (±0.254) | 62.57% | 84.18% |
| Led | Pearson | 1.000 (±0.000) | 100.00% | 100.00% |
| | Spearman | 1.000 (±0.000) | 100.00% | 100.00% |
| Led (sampled) | Pearson | 0.990 (±0.009) | 100.00% | 100.00% |
| | Spearman | 0.988 (±0.017) | 100.00% | 100.00% |
| Digits | Pearson | 0.881 (±0.047) | 38.01% | 98.39% |
| | Spearman | 0.891 (±0.041) | 44.80% | 99.55% |

## G  Notations, and definitions of entropies and mutual information

As a minimal introduction to information theory, we recall in this section several definitions from information theory (see Cover and Thomas [2012], for further properties). The presentation below is largely based on the Supplementary material of [Louppe et al., 2013], which is reproduced here for the convenience of the reader.

We suppose that we are given a probability space $(\Omega, \mathcal{E}, \mathbb{P})$ and consider random variables defined on it taking a finite number of possible values. We use upper case letters to denote such random variables (e.g. $X, Y, Z, W \ldots$) and calligraphic letters (e.g. $\mathcal{X}, \mathcal{Y}, \mathcal{Z}, \mathcal{W} \ldots$) to denote their image sets (of finite cardinality), and lower case letters (e.g. $x, y, z, w \ldots$) to denote one of their possible values. For a (finite) set of (finite) random variables $X = \{X_1, \ldots, X_i\}$, we denote by $P_X(x) = P_X(x_1, \ldots, x_i)$ the probability $\mathbb{P}(\{\omega \in \Omega \mid \forall \ell : 1, \ldots, i : X_\ell(\omega) = x_\ell\})$, and by $\mathcal{X} = \mathcal{X}_1 \times \cdots \times \mathcal{X}_i$ the set of joint configurations of these random variables. Given two sets of random variables, $X = \{X_1, \ldots, X_i\}$ and $Y = \{Y_1, \ldots, Y_j\}$, we denote by $P_{X|Y}(x \mid y) = P_{X,Y}(x, y)/P_Y(y)$ the conditional density of $X$ with respect to $Y$.[3]

With these notations, the joint (Shannon) entropy of a set of random variables $X = \{X_1, \ldots, X_i\}$ is thus defined by

$$H(X) = -\sum_{x \in \mathcal{X}} P_X(x) \log_2 P_X(x),$$

---

[3]To avoid problems, we suppose that all probabilities are strictly positive, without fundamental limitation.

while the mean conditional entropy of a set of random variables $X = \{X_1, \ldots, X_i\}$, given the values of another set of random variables $Y = \{Y_1, \ldots, Y_j\}$ is defined by

$$H(X \mid Y) = -\sum_{x \in \mathcal{X}} \sum_{y \in \mathcal{Y}} P_{X,Y}(x, y) \log_2 P_{X \mid Y}(x \mid y).$$

The mutual information among the set of random variables $X = \{X_1, \ldots, X_i\}$ and the set of random variables $Y = \{Y_1, \ldots, Y_j\}$ is defined by

$$
\begin{aligned}
I(X; Y) &= -\sum_{x \in \mathcal{X}} \sum_{y \in \mathcal{Y}} P_{X,Y}(x, y) \log_2 \frac{P_X(x) P_Y(y)}{P_{X,Y}(x, y)} \\
&= H(X) - H(X \mid Y) \\
&= H(Y) - H(Y \mid X).
\end{aligned}
$$

The mean conditional mutual information among the set of random variables $X = \{X_1, \ldots, X_k\}$ and the set of random variables $Y = \{Y_1, \ldots, Y_j\}$, given the values of a third set of random variables $Z = \{Z_1, \ldots, Z_i\}$, is defined by

$$
\begin{aligned}
I(X; Y \mid Z) &= H(X \mid Z) - H(X \mid Y, Z) \\
&= H(Y \mid Z) - H(Y \mid X, Z) \\
&= -\sum_{x \in \mathcal{X}} \sum_{y \in \mathcal{Y}} \sum_{z \in \mathcal{Z}} P_{X,Y,Z}(x, y, z) \log_2 \frac{P_{X \mid Z}(x \mid z) P_{Y \mid Z}(y \mid z)}{P_{X,Y \mid Z}(x, y \mid z)}.
\end{aligned}
$$

We also recall the chaining rule

$$I(X, Z; Y \mid W) = I(X; Y \mid W) + I(Z; Y \mid W, X),$$

and the symmetry of the (conditional) mutual information among sets of random variables

$$I(X; Y \mid Z) = I(Y; X \mid Z).$$


## References

Thomas M Cover and Joy A Thomas. Elements of Information Theory. John Wiley & Sons, 2012.