# OpenReview forum: "From global to local MDI variable importances for random forests and when they are Shapley values"
_NeurIPS.cc/2021/Conference — NeurIPS 2021 Poster_

### Official Review · Reviewer_5L4p · 2021-07-15

**Rating:** 6
**Confidence:** 5

**Summary:**

The authors provide conditions under which tree-based mean decrease in impurity (MDI) measures are Shapley values. This is an interesting and potentially useful connection, with particular implications for computation time of Shapley values in certain settings.


**Limitations And Societal Impact:**

No negative impacts are discussed, but as in all cases in explainable AI there is potential for results to be misused; this should be discussed.


**Main Review:**

- More discussion of totally randomized trees is necessary: when are these reasonable or feasible to compute? What happens in finite samples?
- Local MDI has a strange interpretation; the local MDI for an observation appears to depend on other instances, and thus is not truly local. Please provide more discussion of this measure and when it has a useful interpretation.
- Does the summation from local to global MDI in Equation 9 hold? It seems that local MDI can be negative, but global MDI cannot.
- I would redefine "asymptotic local MDI" as "future local MDI", since the term "asymptotic" doesn't really fit for local importance
- Please define notation (e.g., $I$ and $H$) prior to its use
- In the related work section, please discuss totally randomized trees in more detail
- The figures are difficult to read, both due to their small size and the information contained. Please consider a different format (e.g., a table of values to replace Figure 2) to facilitate easier comparisons, and please increase the font size in all cases.
- Please include more discussion of how to handle continuous variables in this framework
- Please consider editing the section headings to make the manuscript flow more logically, and check for typos and other formatting
- The proof of Theorem 3 provided in the supplement is too sparse: only an if-then relation (not an if and only if) appears to be proved there.

**Time Spent Reviewing:**

4

---

> ### Author Response · Authors · 2021-08-10
> **Response to Reviewer 5L4p**
>
> We thank the reviewer for his/her positive assessment of our paper.
> ***
> >C3-0: More discussion of totally randomized trees is necessary: when are these reasonable or feasible to compute? What happens in finite samples?
>
> R3-0: The assumption of totally randomized trees appeared in prior works to be necessary to show the consistency of the global MDI with respect to feature relevance as a way of decoupling the tree building from the data. In practice (finite samples), totally randomized trees are not used when model performances are targeted since non-totally randomized trees would preferably focus on the most useful/relevant features and avoid (random) splits on useless/noisy features. However, empirical works show that non-totally randomized trees may miss some masked features and it is theoretically clear since Louppe et al. (2013)  that the only way of finding all relevant features is to derive the importances from an ensemble of totally randomized trees. In his follow-up work, Louppe (2014) also discusses the finite sample size case where the impurity estimation would degrade in deeper nodes if the sample size is not large enough. Overall, we agree that more discussion/theoretical characterisation of MDI in the finite sample case would be of great interest and definitely deserved to be tackled in future works.
>
> >C3-1: Local MDI has a strange interpretation; the local MDI for an observation appears to depend on other instances, and thus is not truly local. Please provide more discussion of this measure and when it has a useful interpretation.
>
> R3-1: We are not sure to understand what you mean by “truly local”. Local MDI computes the importance of each feature value of an observation from a pre-existing tree-based model. The tree-based model is indeed built using all (training) instances, and in that way, the reviewer is thus right that the local MDI importance scores depend on other instances than the observation itself. However, local MDI collects impurity reductions along the unique path in each tree followed by the observation (to reach the prediction) and therefore for this reason, it is local. Local MDI importance scores are different from one observation to another. Note that this feature is common to all local explanation methods. They all start from a model trained using all training instances.
>
> >C3-2: Does the summation from local to global MDI in Equation 9 hold? It seems that local MDI can be negative, but global MDI cannot.
>
> R3-2: Yes. Indeed local MDI may be negative while global MDI cannot, in the same way that the difference of conditional entropies (inner part of local MDI of Eqn (10)) may be negative but the mutual information (inner part of global MDI of Eqn (4), i.e., a suitably weighted average of differences of conditional entropies) can not.
>
> >C3-3: I would redefine "asymptotic local MDI" as "future local MDI", since the term "asymptotic" doesn't really fit for local importance
>
> R3-3: We do not get that point. Could you please clarify why you would prefer to use “future” instead of “asymptotic”? In our mind, we refer to asymptotic local MDI to stress that the ensemble of randomized trees is built under asymptotic conditions (infinite sample size) and therefore impurity decreases are not longer estimates of (but exactly equal to) Shannon entropies and that all orderings are considered (infinite number of trees). We feel therefore that “asymptotic” is the appropriate term.
>
> >C3-4: Please define notation (e.g.,  I and H) prior to its use
>
> R3-4: We thank the reviewer. H (Shannon entropy) can surely be clarified before its use.
>
> >C3-5: In the related work section, please discuss totally randomized trees in more detail
>
> R3-5: Which aspect would you like to be discussed in more detail? Totally randomized trees are simply an ensemble of randomized trees with the value for the randomization parameter K set to 1. As explained in R3-0,  this has been used in several prior works. We realise however that we have not formally defined them in the paper. In the next iteration of the paper, we will define more explicitly what we mean by totally randomized trees in the background section.
>
> >C3-6: The figures are difficult to read, both due to their small size and the information contained. Please consider a different format (e.g., a table of values to replace Figure 2) to facilitate easier comparisons, and please increase the font size in all cases.
>
> R3-6: We thank the Reviewer for this suggestion. We will take it into account when presenting these results and in future iterations of this work.
>
> >C3-7: Please include more discussion of how to handle continuous variables in this framework
>
> R3-7: Appendix F generalizes local MDI to regression (continuous output variable) and a better characterisation in this context is left as future works. Note that input continuous variables can be handled in this framework.   Splits are then of the form X<= x rather than X=x and local MDI of a continuous variable can be computed directly using (8). Some datasets used in the experiments contain actually continuous variables. We will make this more clear in the paper. Note however that adapting our theoretical analysis to continuous input variables is not trivial and, as mentioned in the conclusion, is left as future work at this stage.
>
> >C3-8: The proof of Theorem 3 provided in the supplement is too sparse: only an if-then relation (not an if and only if) appears to be proved there.
>
> R3-8: We will rewrite the proof to make it more clear. Actually, the proof boils down to the equivalence (if and only if) of the irrelevance property and the set of local irrelevance properties.  In other words, irrelevance already considers all samples x while local irrelevance only considers one x. If local irrelevance is satisfied for all x, then irrelevance is satisfied. The other way around is trivial.
> ***
> We hope that these answers clarify the reviewer's concerns in particular about the use of totally randomized trees (both reasonable and feasible) and continuous variables, and that the reviewer may be even more convinced of the goodness of the paper.

---

> > ### Comment · Reviewer_5L4p · 2021-08-17
> > **Response to authors**
> >
> > Thank you for your thoughtful responses. I have some replies below.
> >
> > R3-0: Your response raises a further question: if totally randomized trees aren't used in finite samples due to worse prediction performance than conventional trees, why should we use them for variable importance? In some sense, this procedure advocates for fitting two distinct procedures in a prediction problem: a conventional tree to do the predicting, and a totally randomized tree for variable importance. This is disconcerting, since the goal of algorithmic feature importance is to explain the *fitted algorithm*.
> >
> > R3-5: I think your comment here would be sufficient -- I wanted a more thorough background on totally randomized trees in this manuscript, since they form the backbone of the paper.

---

> > > ### Author Response · Authors · 2021-08-19
> > > **Response to Reviewer 5L4p**
> > >
> > > Thank you for continuing the discussion and your comments. We hope that our answer will also clarify this additional question.
> > >
> > > ------
> > > R3-0: You are right. We can follow two distinct procedures to compute feature importances, but each approach's motivation and sought goal are different. Feature importances can indeed be used to explain a fitted predictive model whose hyper-parameters (including K) have been tuned for predictive performance. In this case, feature importances would measure to which extent features are useful in this tree-based model. In particular, if some features are not used (because K > 1), then they would be marked as not important. However, in critical applications as in science or medicine, one could wish to uncover and understand exhaustively all the interactions and relations between the inputs and the output. For example, in genomics, we would be interested in identifying all genes associated with a disease. Using totally randomized trees would uncover all those interactions, while non-totally randomized may not, since a small subset of features may be sufficient for good predictive performance. Actually, the goal in the latter case would not be to explain a fitted model but to highlight intrinsic properties of the data distribution. This is what we discuss in the last paragraph of Section 5.
> > >
> > > R3-5: Thank you for the clarification, we will then add this comment to the future iteration of the paper.

---

### Official Review · Reviewer_gLH8 · 2021-07-16

**Rating:** 7
**Confidence:** 4

**Summary:**

The paper’s contributions are on variable importance measures in tree and tree ensembles. In particular, the paper provides an interpretation of the “mean decrease of impurity” measure (MDI) of totally randomised trees as Shapley values where the characteristic function of the underlying game is the mutual information between a feature variable set and the target variable. Moreover, a novel local variant of MDI measure is introduced that quantifies the impact of variable for the classification of a specific input point, which can again be interpreted as Shapley values in the case of totally randomised trees.

**Limitations And Societal Impact:**

The paper should have a positive societal impact because its contributions help to make machine learning more interpretable.

**Main Review:**

Given their availability in popular ML packages, MDI is widely used in practice to assess variable importance. However, its interpretation is not straightforward. Hence, results are important that characterise MDI in terms of other known quantities. The given results strongly build on previous work that connected MDI to mutual information. While the step from there to Shapley values is relatively small, it is still an important novel insight. Moreover, the proposed local variant of MDI is appealing for its simplicity and natural connection to global MDI which can be decomposed as an average over all local MDIs. Overall, the paper is a pleasantly well-written and provides a useful collection of results that help to interpret the classification of randomised tree ensembles.

Some minor suggestions for improvement:
- The experiments are nice to illustrate the measures, but perhaps some stronger experiments (potentially with synthetic data) could be devised that demonstrates that MDI can become misleading for large K when it is no longer a Shapely value
- The paper manages to be mostly self-contained despite the relatively wide set of concepts it needs to introduce. What feels is missing, is a minimal introduction to mutual information and its properties as they are essential for the main results.
- In the definition of strong monotonicity, the index of the MC function should be m
- In Definition 2, it seems more readable to avoid the all quantifier over labels and simply require equality of the two conditional distributions of Y

**Time Spent Reviewing:**

6

---

> ### Author Response · Authors · 2021-08-10
> **Response to Reviewer gLH8**
>
> We thank the reviewer for his/her very positive assessment of our paper.
> ***
> > C2-0: Given their availability in popular ML packages, MDI is widely used in practice to assess variable importance. However, its interpretation is not straightforward. Hence, results are important that characterise MDI in terms of other known quantities. The given results strongly build on previous work that connected MDI to mutual information. While the step from there to Shapley values is relatively small, it is still an important novel insight. Moreover, the proposed local variant of MDI is appealing for its simplicity and natural connection to global MDI which can be decomposed as an average over all local MDIs.
>
> > C2-1: Overall, the paper is a pleasantly well-written and provides a useful collection of results that help to interpret the classification of randomised tree ensembles.
>
> R2-0 and R2-1: We thank the reviewer for these comments. We also believe that one key contribution of this paper is the link between global and local MDI importance measures and hope that this work will contribute to a better interpretation of randomized tree ensembles.
>
> >C2-2: The experiments are nice to illustrate the measures, but perhaps some stronger experiments (potentially with synthetic data) could be devised that demonstrates that MDI can become misleading for large K when it is no longer a Shapely value
>
> R2-2: We agree with the reviewer that some stronger experiments showing the impact of K could be an interesting addition to the paper and we will think about it. However we believe that the literature (see e.g. Louppe et al. (2013) and Sutera et al. (2018) cited in the main paper in Section 3.2) already well covered this aspect of tree-based importance measures. We also hope that knowing that MDI measure for large K departs from the Shapley value assumption of equal treatment features (and combinations) could already provide a good insight of the kind of results such experiments would confirm.
>
> >C2-3: The paper manages to be mostly self-contained despite the relatively wide set of concepts it needs to introduce. What feels is missing, is a minimal introduction to mutual information and its properties as they are essential for the main results.
>
> R2-3: We will surely take this remark into account when correcting the paper, and this can surely be added in supplementary materials.
>
> >C2-4: In the definition of strong monotonicity, the index of the MC function should be m
>
> R2-4: We thank the reviewer for pointing this out. This will be corrected.
>
> >C2-5: In Definition 2, it seems more readable to avoid the all quantifier over labels and simply require equality of the two conditional distributions of Y
>
> R2-5: Thanks. We will surely take this remark into account to improve the readability of our paper.
> ***
>  We hope that our answers will consolidate the reviewer's initial review.

---

### Official Review · Reviewer_TfJb · 2021-07-29

**Rating:** 6
**Confidence:** 5

**Summary:**

This paper demonstrates a connection between Shapley values and a tree-based measure of global feature importance, then proposes a local version of that tree-based feature importance measure. The paper also offers a discussion of the properties satisfied by the proposed method and runs empirical tests of feature importance methods on a few datasets.

**Limitations And Societal Impact:**

The authors adequately addressed potential negative societal impact of their work.

**Main Review:**

I would rank the overall originality and quality of the current form of this paper rather low. While methods for identifying important features in tree-based models are generally of great interest due to the wide-spread use of these types of models in deployment, the significance of the proposed method is not well demonstrated (either theoretically or empirically), and several of the contributions of the paper have been discussed already in prior works.

One of the major contributions of this paper is the connection of MDI variable importance scores to Shapley values under certain conditions (randomized trees under asymptotic conditions). While developing the understanding of connections between older, more established tree-based methods for variable importance and more recent Shapley value-based methods is certainly of value and of interest, the connection between these methods has already been discussed in prior work, and in my opinion has been explained in a more intuitive way there.

Specifically, the method discussed in this paper can be thought of as a modified version of the Saabas algorithm [1], where output differences are replaced by impurity reductions (see lines 296-297 in this paper). The connection between algorithms like Saabas/MDI and the Shapley value was already made in Lundberg et al.’s 2019 paper (see the bottom of pg. 20-21 in the arxiv version of that paper) – Saabas/MDI only consider a single ordering to add players to the coalition, while the Shapley value averages over all orderings. When there are randomized trees under asymptotic conditions, all possible feature orderings will be considered in equal proportions, making these equivalent [2].

Furthermore, this paper incorrectly states that TreeSHAP only decomposes model predictions. A significant portion of that work involves an algorithm called “Independent Tree SHAP” (see methods 10.2 of [2]), that allows for the attribution of a model’s loss to individual features at the local level, which is also discussed in [3]. TreeSHAP loss attributions should certainly be more thoroughly discussed theoretically or at least compared to empirically.

I think there are a variety of avenues that could make this work significantly more impactful – either a theoretical analysis outside of the asymptotic conditions, or empirical results for more representative datasets (i.e. not toy examples like the digits/LED used in this paper) that the MDI values were of similar quality to SHAP while being meaningfully faster on these more realistic datasets.

As a minor comment, the notation is pretty dense, confusing, and occasionally overloaded – leading to ambiguity (like $v$ referring to both the characteristic function of a TU game [see line 48] and the variable tested in a certain node [see line 98]).

[1] Ando Saabas. Interpreting random forests. 2014. http://blog.datadive.net/interpreting-random-forests/
[2] Lundberg, Scott M., et al. "From local explanations to global understanding with explainable AI for trees." https://arxiv.org/abs/1905.04610
[3] Covert et al. "Understanding global feature contributions with additive importance measures." https://arxiv.org/abs/2004.00668


**Time Spent Reviewing:**

7 hours

---

> ### Author Response · Authors · 2021-08-10
> **Response to Reviewer TfJb**
>
> We thank the reviewer for his remarks and the time spent on this review.
> ***
> >C1-1: I would rank the overall originality and quality of the current form of this paper rather low. While methods for identifying important features in tree-based models are generally of great interest due to the wide-spread use of these types of models in deployment, the significance of the proposed method is not well demonstrated (either theoretically or empirically), and several of the contributions of the paper have been discussed already in prior works.
>
> >One of the major contributions of this paper is the connection of MDI variable importance scores to Shapley values under certain conditions (randomized trees under asymptotic conditions). While developing the understanding of connections between older, more established tree-based methods for variable importance and more recent Shapley value-based methods is certainly of value and of interest, the connection between these methods has already been discussed in prior work, and in my opinion has been explained in a more intuitive way there.
>
> >Specifically, the method discussed in this paper can be thought of as a modified version of the Saabas algorithm [1], where output differences are replaced by impurity reductions (see lines 296-297 in this paper). The connection between algorithms like Saabas/MDI and the Shapley value was already made in Lundberg et al.’s 2019 paper (see the bottom of pg. 20-21 in the arxiv version of that paper) – Saabas/MDI only consider a single ordering to add players to the coalition, while the Shapley value averages over all orderings. When there are randomized trees under asymptotic conditions, all possible feature orderings will be considered in equal proportions, making these equivalent [2].
>
> R1-1: Our work is indeed related to several works (all discussed in Section 5) but we kindly disagree with the reviewer that our contributions would not be original.
>
> While the reviewer links MDI and Saabas, it's important to note that prior to our work, only the standard **global** MDI measure existed, while Saabas is a local measure. The first contribution of our paper (**Section 3**) is to show that **global** MDI importances are Shapley values with respect to mutual information in specific conditions. In [2] (page 20-21), the authors claim that Louppe (reference [40] in [2]) has shown that (global) MDI was consistent (strongly monotone in our paper) in the same conditions but to the best of our knowledge, there is no such claim/proof in [40]. Theorem in [2] (pages 20-21) is about Saabas' measure, which is local and decomposes model predictions, not global MDI. So, we believe that results in Section 3 (theorem 1, as well as the discussion of the case of non totally randomized trees) are original (although they derive directly from decomposition (4), which comes from (Louppe et al., 2013)).
>
> Our second contribution is the **local** MDI importance measure in **Section 4**. This measure derives naturally from global MDI. It turns out that it indeed is simply Saabas' measure with output differences replaced by impurity reductions (as clearly acknowledged in Section 5 of our paper) but it was designed to be a natural local version of global MDI, not as a modification of Saabas' mesure. We see the link between local MDI and Saabas as a nice result that actually improves our understanding of Saabas' measure, which has originally been proposed as a heuristic. To the best of our knowledge, nobody has proposed or studied this specific version of Saabas' measure and has shown its link with global MDI. Theorem 2 is indeed close to Theorem 1 on page 20/21 in the appendices of [2] (we highlight this link in Section 5) but our proof is however more formal than the proof in [2] and also specific to our setting where mutual informations are decomposed. We thus believe it is original. Finally, using mutual information allows us also to link local MDI with a newly introduced notion of local relevance (Section 4.4), which is only possible for local MDI.
>
> One important side contribution of our paper (Sections 3.2, 5 and 6) is to highlight the link between the properties of importance measure and the way the forest is constructed (the randomization level K). We believe this link was an oversight in the existing literature, where forests were trained with default settings only and considered as black-box models.
>
> For all these reasons, we believe our work is mostly original and that it fills several missing gaps in the literature.
>
> >C1-2: Furthermore, this paper incorrectly states that TreeSHAP only decomposes model predictions. A significant portion of that work involves an algorithm called “Independent Tree SHAP” (see methods 10.2 of [2]), that allows for the attribution of a model’s loss to individual features at the local level, which is also discussed in [3]. TreeSHAP loss attributions should certainly be more thoroughly discussed theoretically or at least compared to empirically.
>
> R1-2: We agree with the reviewer that (independent) TreeShap can decompose model's loss, at a higher computational cost however. We felt nevertheless that the authors of TreeShap advocate the decomposition of model's prediction as the way to go, since local accuracy is mentioned as a desirable feature of explanation methods and most experiments in the TreeShap paper are focused on this decomposition. We will however modulate our claim in Section 5 and more thoroughly discuss TreeShap loss attributions. The link with [3] that is indeed focused on loss attribution (but is not tree specific) is already clearly highlighted in Section 5.
>
> >C1-3: I think there are a variety of avenues that could make this work significantly more impactful – either a theoretical analysis outside of the asymptotic conditions, or empirical results for more representative datasets (i.e. not toy examples like the digits/LED used in this paper) that the MDI values were of similar quality to SHAP while being meaningfully faster on these more realistic datasets.
>
> R1-3: We agree that such contributions would be highly valuable and we plan to work on that as future work. Concerning the empirical results, while we focus on digits and LED in the main paper because importance scores are easy to interpret visually on these datasets, note that experiments on four additional datasets are already provided in the Supplementary materials that confirm the results on digits and LED. Note also that the computational advantage of global/local MDI importance scores is obvious since these methods have virtually no additional cost with respect to making a prediction with the forest.
>
> >C1-4: As a minor comment, the notation is pretty dense, confusing, and occasionally overloaded – leading to ambiguity (like v referring to both the characteristic function of a TU game [see line 48] and the variable tested in a certain node [see line 98]).
>
> R1-4: We thank the reviewer for pointing this out. We will surely take this remark into account to improve the paper.
>
>
> ***
>
> In light of our efforts for answering all the elements raised in the review, we hope our answers will have resolved most of the reviewer’s concerns about the originality and quality of our work and that the initial rating will be revised in a fair and factual manner.

---

> > ### Comment · Reviewer_TfJb · 2021-08-21
> > **Response to author comments**
> >
> > Thank you for your response.
> >
> > To address your comments in the first part of (R1-1:), I'd say that it's possible I undervalued the contribution of connecting global MDI to Shapley values, and I'd be happy to revise my initial rating. I certainly agree with you that the "proof" shown by the authors of [2] seems very informal and that more throughly explicating this and showing that it is correct is a valuable contribution. The issue that I was trying to raise with your contribution in Section 3 is that while it directly follows from Louppe, it would be much more useful to help readers understand _why_ the values are Shapley values beyond simply showing that the coefficients match. Even if this would be recapitulating info from [2] or from the Louppe paper, if your contribution has to do with helping connect disparate elements of the literature, doing so in a way that helps deepen readers' understanding seems important. Still, again, I'm happy to raise my score a bit higher.
> >
> > To address your comments in the second part of (R1-1:), I appreciate that the second part of your contribution is "the local MDI importance measure in Section 4," which I noted in my summary of your paper, where I said it "proposes a local version of that tree-based feature importance measure." This is what motivated my comments in (C1-3:), where I said that "a theoretical analysis outside of the asymptotic conditions, or empirical results for more representative datasets" would be important. If your contribution is a novel feature importance measure, I think it's important to show that it's actually useful in some way -- I disagree that this should be left to future work. Since prior work on feature attribution has shown that measures like Saabas' measure or MDI can be inconsistent in even very simple cases, showing that on some practical, non-toy dataset the attributions end up being similar to Shapley values while being faster to calculate would be a very convincing demonstration of the value of this local MDI importance measure.

---

> > > ### Author Response · Authors · 2021-08-29
> > > **Second response to reviewer TfJb**
> > >
> > > Thank you for your response.
> > >
> > > > First part of (R1-1:):
> > >
> > > We are happy that you re-evaluated the contribution of connecting global MDI with shapley values. We agree that it would be useful to explain more intuitively why global MDI importances are shapley values (when K=1) to support the theoretical results and we will do so in the next revision of our paper.
> > >
> > > > Second part of (R1-1:):
> > >
> > > Obviously, more experiments are always better and it would not be difficult to repeat the same experiments with more datasets of larger sizes.
> > >
> > > The focus of the paper is however on the theoretical results, which we believe already constitute a significant contribution per se and provide solid arguments in favor of the local MDI scores: they decompose naturally global MDI scores (whose empirical relevance has been highlighted in many papers), their computational cost is negligible, they are shown to satisfy the properties of Shapley values when hyper-parameters are tuned appropriately (4.3), and they can be linked to a new statistical relevance property (Section 4.4). We also provide experiments on 6 datasets, where local MDI scores are shown to be very close to other related more costly measures. All in all, we believe that these results already represent enough material for a 9-pages conference paper and we are looking forward to extending them with more empirical results for a follow-up journal publication.
> > >
> > > Also, we disagree that we are using only toy datasets with respect to standards in the related literature. Again, we choose to illustrate our approach with two toy examples in the main paper but we provide results with four additional datasets of greater complexity in the supplementary material (more features with either integer, real, or mixed integer/real values). Note that other papers on similar topics have considered either the same datasets or datasets of similar complexity. For example, in the SAGE paper (Covert, Lundberg and Lee, NeurIPS 2020), the authors use the breast cancer datasets for logistic regression and smaller datasets (<=20) for tree-based models. Applying the methods on a real large-scale problem where important features can be evaluated by domain experts would be very valuable but is clearly out of the scope of this paper. We finally believe that showing that one measure is better than another is not trivial, precisely because our theoretical results show that they are tightly linked.
> > >
> > > About this comment:
> > >
> > > > Since prior work on feature attribution has shown that measures like Saabas' measure or MDI can be inconsistent in even very simple cases,
> > >
> > > We specifically address this by showing conditions in which global and local MDI measures are consistent. Note that MDI was shown to be inconsistent in [2] when computed from two fixed trees irrespectively of the way these trees are constructed from the data. We believe this is not appropriate as MDI is not a model agnostic metric; it is tightly linked to the way the trees are built (the value of K). If you fix K, it’s not easy to find two data distributions that highlight inconsistency of global MDI scores obtained from forests built with this value of K from the two distributions. Actually, it took us a lot of effort to come up with the example in Appendix B.
> > >
> > > Note also that even though other measures such as SAGE and TreeShap are consistent, they are actually consistent with respect to the random forest model they are explaining, not with respect to the (unknown) ground truth model. We show clearly in our experiments (Figure 1) that SAGE and TreeSHAP are impacted as much as MDI by the randomization parameter K. This is discussed in Section 5. Despite the fact that they are consistent, SAGE and TreeSHAP are tied to the quality of the model they explain. Being consistent with respect to a model does not mean being consistent with respect to the ground truth that generated the data from the model was trained.

---

### Decision · Program_Chairs · 2021-09-27

**Decision:**

Accept (Poster)

**Comment:**

This paper presents previously unknown connection between the global Mean Decrease of Impurity (MDI) variable importance scores and Shapley values under certain conditions. The authors also derive local MDI feature importance measure link with Shepley values.  The paper presents novel insights, and the local MDIs are likely practically useful for its simplicity. This paper is clearly written and structured well. Overall, this paper constitutes an important contribution to the field and passes the bar for the acceptance to NeurIPS.